# GAN-enhanced machine learning and metabolic modeling identify reprogramming in pancreatic cancer

Tahereh Razmpour[1], Masoud Tabibian[1], Arman Roohi[2], Rajib Saha[1]*

**1** Department of Chemical and Biomolecular Engineering, University of Nebraska-Lincoln, Lincoln, Nebraska, United States of America, **2** Department of Electrical and Computer Engineering, University of Illinois, Chicago, Chicago, Illinois, United States of America

* rsaha2@unl.edu

## Abstract

Pancreatic ductal adenocarcinoma is one of the deadliest forms of cancer, presenting significant clinical challenges due to poor prognosis and limited treatment options. Understanding the metabolic reprogramming that drives this disease is crucial for identifying new therapeutic targets and improving patient outcomes. We developed a novel computational framework integrating genome-scale metabolic modeling with machine learning to identify metabolic signatures and therapeutic vulnerabilities in pancreatic cancer. To address the inherent class imbalance in cancer datasets, we generated synthetic healthy samples using a Wasserstein Generative Adversarial Network with Gradient Penalty, implementing a rigorous three-step biological filtration process to ensure their validity. This approach enabled the creation of a balanced dataset for robust comparison of healthy versus cancerous metabolic states. Our machine learning classifier achieved 94.83% accuracy in distinguishing between these states, demonstrating the effectiveness of our integrated approach. Systems-level analysis revealed three key dysregulated pathways: heparan sulfate degradation, O-glycan metabolism, and heme degradation. We identified impaired lysosomal degradation of heparan sulfate proteoglycans as a potential contributor to disease pathogenesis, providing a mechanistic explanation for the previously observed association between lysosomal storage disorders and pancreatic cancer. Additionally, nervonic acid transport emerged as the most discriminative reaction between healthy and cancerous states, with gene-level analysis highlighting fatty acid binding proteins, fatty acid transporters, and acyl-CoA synthetases as key molecular drivers of metabolic reprogramming. Our multi-level approach connected genetic drivers to functional metabolic consequences, revealing coordinated upregulation of fatty acid transport and activation processes. These findings enhance our understanding of pancreatic cancer metabolism and present potential therapeutic targets, demonstrating the value of integrated computational approaches in cancer research.

**Data availability statement:** All the codes and materials used for this study are available at https://github.com/ssbio/PDAC.

**Funding:** This study was supported by an NIGMS MIRA Award 5R35GM143009 to RS. The funders had no role in study design, data collection and analysis, decision to publish, or preparation of the manuscript.

**Competing interests:** The authors have declared that no competing interests exist.

## Author summary

Pancreatic cancer is one of the deadliest forms of cancer, with most patients surviving less than five years after diagnosis. One major challenge in studying this disease is understanding how cancer cells rewire their metabolism—the chemical processes that provide energy and building blocks for growth. We developed a new computational approach that combines two powerful tools: computer models that simulate cellular metabolism and artificial intelligence techniques that can identify patterns in large datasets. Our method addresses a common problem in cancer research: there are far more cancer samples available for study than healthy tissue samples. We used an artificial intelligence technique called generative adversarial networks to create realistic synthetic healthy samples, allowing us to build a balanced dataset for comparison. We then applied machine learning to identify which metabolic processes are most altered in pancreatic cancer. Our analysis revealed three key metabolic pathways that are disrupted in pancreatic cancer: processes involved in breaking down cellular components, modifying proteins with sugar molecules, and processing waste products. We also discovered that cancer cells dramatically increase their uptake of specific fatty acids, which they need for rapid growth and survival in harsh tumor environments. These findings provide new insights into how pancreatic cancer cells adapt their metabolism and suggest potential targets for developing new treatments.

## Introduction

Pancreatic Ductal Adenocarcinoma (PDAC) is a disease with poor prognosis and a highly aggressive form of cancer, largely due to late-stage diagnosis and limited treatment options. The majority of PDAC patients (80–85%) present with locally advanced or metastatic disease at diagnosis, when curative surgical resection is no longer feasible [1]. This delayed detection dramatically impacts survival, as 5-year survival rates decrease from approximately 32% for localized disease to only 3% for metastatic disease [2]. The 5-year survival rate for PDAC remains below 10%, emphasizing the critical need for improved early detection methods and personalized treatment strategies [3]. Given the complex nature of PDAC and its large variability in patients and symptoms, there is a growing interest in leveraging advanced computational approaches to enhance diagnostic accuracy and treatment efficacy. These approaches can be categorized into two groups: data-driven approaches [4] and first-principles/metabolic model-driven approaches [5]. The metabolic model-driven approaches include integrating patient-specific transcriptomics data using genome-scale metabolic modeling (GSM) to enhance early detection [6]. Additionally, in the data-driven approaches, machine learning algorithms such as one-dimensional convolutional neural networks are used to analyze RNASeq data for the detection of various cancer types, with feature selection techniques being employed to enhance prediction accuracy [6,7].

Genome-scale metabolic modeling (GSM) has emerged as a powerful tool for understanding the metabolic repro-gramming that occurs in a wide range of human diseases including PDAC. A GSM can represent gene-protein-reaction associations, thus representing the entire metabolic landscape of a human cell-type of interest. By overlaying omics data (e.g., transcriptomics or proteomics) via switch-based or valve-based approach onto the global human metabolic model, Human1 [8], such cell-specific metabolic models can be developed for the prediction of metabolic vulnerabilities [9]. While the switch-based algorithms (e.g., iMAT [10] or tINIT [11]) remove inactive or lowly expressed genes by setting the corresponding reaction boundaries to zero for reconstruction of the context-specific models, the valve-based algorithms (e.g., E-flux [12], SPOT [13], and EXTREAM [14]) adjust the upper and lower bounds for reactions proportionally to the normalized expression of associated genes, without discretizing gene expression data [15]. Constraint-based modeling is the primary method used to analyze GSMs since it provides a way to make predictions about metabolic behavior without requiring detailed kinetic parameters for every reaction [16,17]. These have been applied to various cancer types such as PDAC [5], liver cancer [18], breast cancer [19], and colorectal cancer [20]. The reconstruction of these cancer-specific GSMs and their subsequent constrained-based analysis aim to identify potential therapeutic targets and metabolic vulner-abilities across different malignancies [21,22]. These studies have revealed alterations in key metabolic pathways, includ-ing glycolysis in PDAC [5] and hepatocellular carcinoma [18], cholesterol biosynthesis in PDAC [5] and colorectal cancer [20], and lipid metabolism in PDAC [5], that contribute to different cancer types progression. Furthermore, these analyses can uncover changes in gene expression patterns associated with these reprogrammed pathways, providing insights into the genetic basis of metabolic alterations in cancer [23]. GSMs, though effective in capturing cellular metabolic land-scapes, often incorporate regulatory mechanisms indirectly by assuming linear relationships between gene expression (or protein abundance) and metabolic activity. These simplifications can introduce infeasibilities and necessitate iterative design–build–test–learn (DBTL) cycles, which are time-consuming and may be prone to human bias [24].

Parallel to the development of metabolic modeling approaches, data-driven methods, particularly machine learning (ML), have gained traction in cancer research. ML algorithms have been employed to analyze complex genomic and transcriptomic datasets, aiming to identify diagnostic and prognostic biomarkers for PDAC. For instance, Sadewo et al. [25] proposed a machine learning approach with the twin support vector machine (TWSVM) method as a new approach to detecting pancreatic cancer early. In addition, Osipov et al. [4] introduced the Molecular Twin, an AI platform that inte-grates multi-omics data to accurately predict disease survival for PDAC patients. Furthermore, another study utilized PCA and random forest modeling to address complex gene-gene interactions, improving the accuracy of cancer driver gene identification in pancreatic cancer, and uncovered new potential therapeutic targets like MXRA5 and NDUFA6, which are linked to poor survival in PDAC patients [26]. It goes without saying that pure ML approaches often focus solely on gene expression data without considering the complex relationships within metabolic networks. This limitation can lead to over-looking important metabolic interactions and regulatory mechanisms [27].

The integration of GSM and ML approaches offers a promising strategy for PDAC research, combining GSM's compre-hensive metabolic representation with ML's data-driven efficiency to overcome limitations of both methods: manual cura-tion and potential human bias in GSM, and the risk of overlooking important metabolic interactions due to a focus on gene expression data in ML [28]. This combination allows for the incorporation of biological knowledge into data-driven models, potentially improving their accuracy and interpretability. Several studies have demonstrated the power of this integrated approach in cancer research. For example, Lee et al. [29] used different switch approaches such as tINIT, GIMME, and rFASTCORMICS for generating the context-specific models of different cancer types. Afterwards, they also used one-dimensional convolutional neural network to classify the patient-specific genome-scale metabolic models to cancer types, based on their reaction contents or flux data. This integrated approach was applied across various cancer types with the objective of assessing the impact of different combinations of model extraction and simulation methods on the biological accuracy of genome-scale metabolic models specific to cancer patients. In a recent study, Tabibian et al. [30] developed a multi-level approach integrating GSM and Random Forest classification to investigate metabolic alterations in lung cancer

and the role of mast cells in the tumor microenvironment. Their analysis revealed selective upregulation of specific amino acids (valine, isoleucine, histidine, and lysine) in the aminoacyl-tRNA pathway and identified significant alterations in mast cell metabolism, including enhanced histamine transport and increased glutamine consumption in cancerous tissues. In addition, Lewis and Kemp [31] utilized flux balance analysis (FBA) to predict metabolite production rates, incorporated them into gradient boosting machine (GBM) classifiers, and identified biomarkers associated with radiation resistance by calculating Shapley Additive Explanations (SHAP) values as feature importances across all cancerous patients.

The integration of GSM and ML approaches, while powerful, introduces new challenges in model development and validation [27]. A critical consideration in the development of ML-informed metabolic models is the balance of the dataset. Imbalanced datasets, in which one class (e.g., cancer samples) significantly outnumbers the other (e.g., healthy samples), can lead to biased predictions and reduced model performance [32]. This issue is particularly pertinent in cancer research, where healthy samples are often underrepresented. Several studies have highlighted the importance of addressing this imbalance through techniques such as oversampling, undersampling, or synthetic data generation [33,34]. Su et al. [35] used a Generative Adversarial Networks (GANs) method, which can synthesize realistic data, to mitigate limited data and class imbalance issues for automatic diagnosis of skin cancer images. In addition, Suresh et al. [36], proposed an enhanced generative adversarial network (E-GAN) to deal with the class imbalance issue for any medical data classification, and they evaluated the performance of it on the Breast Cancer Wisconsin Dataset.

Despite the advances in both GSM and ML approaches, there remains a significant gap in the integrated application of these methods for PDAC diagnosis, particularly in the context of balanced datasets. Most studies have focused on either metabolic modeling [5] or machine learning [4] separately, without fully leveraging the synergistic potential of these approaches. In this work, we address these gaps by developing a novel framework that combines GSM, constraint-based modeling, and ML techniques to identify PDAC-specific metabolic signatures and therapeutic vulnerabilities. Specifically, we employ a switch-based modeling approach rather than valve-based algorithms, as this enables us to focus on the most significant metabolic differences driven by highly expressed genes and reactions rather than capturing incremental variations in low-expression features. Our approach incorporates data balancing techniques to address the inherent imbalance in cancer datasets by generating synthetic data for the minority class and identifies the metabolic reprogramming in presence of PDAC across three different system levels-pathways, reactions, and genes. Moreover, we introduce a novel biological filtration process for synthetic data validation that goes beyond conventional statistical evaluations, ensuring that the generated samples not only match statistical distributions but also adhere to fundamental biological constraints in terms of metabolic functions. Importantly, rather than using machine learning for predictive classification, as is typical in many studies, we apply it as a pattern recognition tool to systematically identify the most distinguishing metabolic features between healthy and cancerous states. Overall, our multi-level analysis represents a significant advancement over existing studies that typically focus on a single level of biological organization, enabling several key advantages: (1) discovery of emergent properties that might be missed in single-level analyses, (2) validation of findings through cross-level consistency, enhancing confidence in identified biomarkers, (3) more precise mechanistic insights connecting genetic drivers to their functional metabolic consequences, and (4) identification of potential intervention points at multiple biological scales, expanding therapeutic possibilities. By integrating these diverse elements, we aim to enhance the accuracy and personalization of PDAC diagnosis, potentially paving the way for improved early detection and targeted therapeutic strategies. A simple schematic of our framework can be seen in Fig 1.

## Results

### Generation and validation of synthetic healthy data using GANs

To address the data imbalance between cancerous (n = 144) and healthy (n = 4) samples in our TCGA dataset, we employed a Wasserstein GAN with Gradient Penalty (WGAN-GP) to generate 251 synthetic healthy gene expression profiles. These synthetic profiles underwent rigorous biological validation through genome-scale metabolic modeling and multi-step filtration processes to ensure their biological relevance.

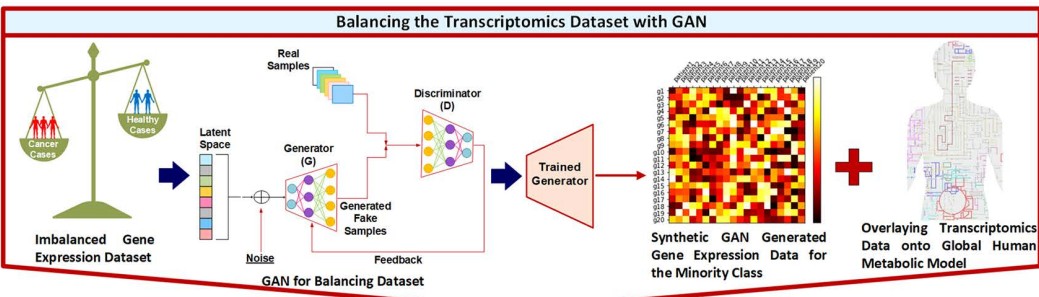

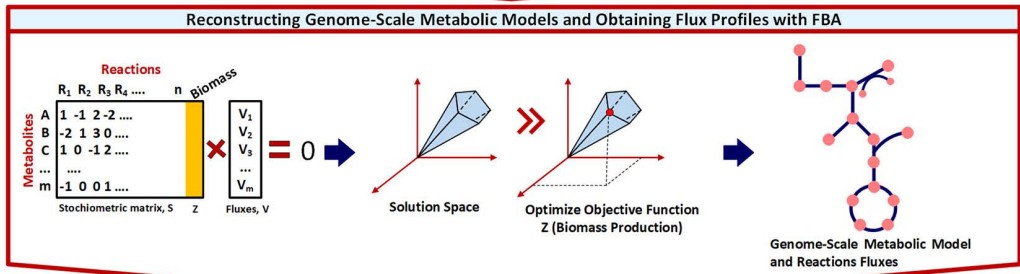

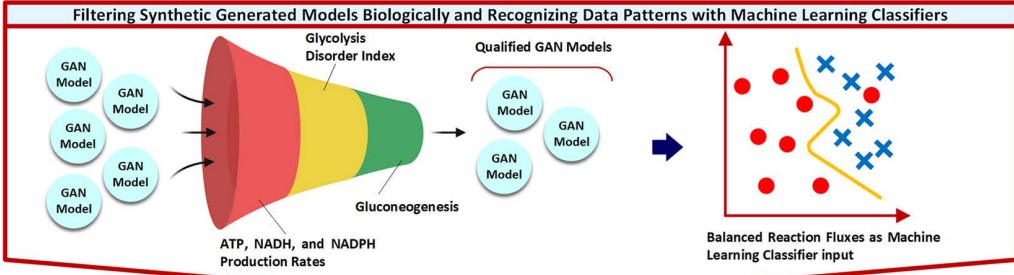

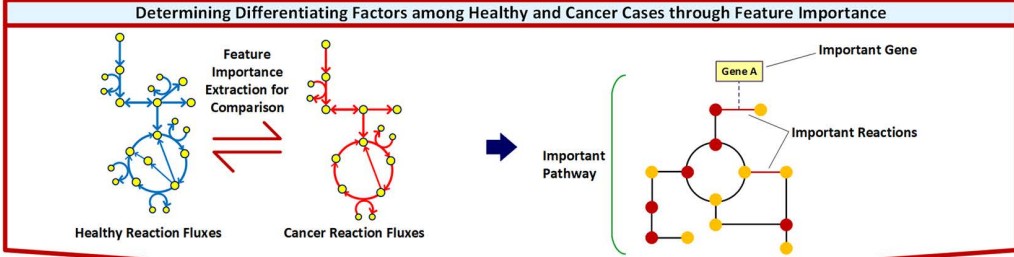

**Pancreatic Cancer Biomarkers**

**Fig 1. Overall workflow for identifying pancreatic cancer biomarkers through balanced machine learning on metabolic models.** The pipeline consists of four main stages: (1) Balancing the transcriptomics dataset with GAN - addressing the imbalance between cancer (red) and healthy (blue) samples by generating synthetic healthy profiles from latent space representation; (2) Reconstruction of genome-scale metabolic models - creating contextualized metabolic models for both sample types using flux balance analysis (FBA); (3) Biological filtration of synthetic models - validating generated healthy models through a three-step process assessing ATP/NADH/NADPH production rates, glycolysis disorder index, and gluconeogenesis activity; and (4) Feature importance analysis - identifying key reactions, pathways, and genes that distinguish between healthy and cancerous metabolic states through machine learning classification. This integrated approach combines transcriptomics data with metabolic modeling to reveal metabolic signatures and identify genes, reactions, and pathways with potential therapeutic relevance in pancreatic cancer. Selected graphical elements created in BioRender. Saha, R. (2025) https://BioRender.com/q0nla2h.

## Genome-scale metabolic model analysis and biological filtration of GAN-generated models

We implemented a comprehensive three-step biological filtration process to evaluate and refine the GAN-generated metabolic models (Fig 2). The first filter assessed the production rates of key metabolites (ATP, NADH, and NADPH), retaining only those models whose production rates fell within the range established by the four original healthy cases. The second filter examined glycolytic function, specifically targeting lactate production in the cytosolic glycolysis pathway, to exclude models exhibiting cancer-like metabolism. The third filter removed models showing gluconeogenesis activity, which is typically absent in well-nourished cells. This rigorous filtration process yielded 140 qualified synthetic healthy models, which, combined with the four original healthy samples, provided a balanced dataset of 144 samples for comparison with the 144 cancerous cases.

The biological relevance of our synthetic data was further validated through t-SNE visualization (S1 Fig) in which demonstrated that the WGAN-GP framework successfully learned the distribution of healthy pancreatic tissue gene expression patterns, as the synthetic healthy gene expressions closely followed the trajectory of the original healthy samples while maintaining clear distinction from cancerous samples.

## Metabolic models classification

Upon creating a balanced dataset comprising of 144 healthy samples for comparison with the 144 cancerous cases, we developed a machine learning approach to distinguish between healthy and cancerous metabolic states. In other words, finding which reactions in the human body are changed hugely in the case of pancreatic cancer compared to the healthy state.

The final dataset used as input for the random forest classifier consisted of 288 samples, including PDAC patients, real healthy cases, and biologically validated synthetic healthy cases generated using WGAN-GP. Additionally, the dataset contained 10,755 features, representing all possible metabolic reactions observed across the models, padded to ensure consistency across samples.

Using the reaction flux profiles obtained from FBA as feature vectors, where each reaction rate represented a distinct feature, we applied the random forest method to classify healthy and cancerous metabolic models. As each metabolic model exhibited a different set of active reactions, we needed to address the challenge of comparing models with varying reaction numbers. We implemented a padding approach where we first created a comprehensive list of all possible reactions observed across our models. For any model that lacked certain reactions from this complete list, we applied padding by assigning NaN values to those missing reactions. This approach allowed us to standardize our dataset while preserving the distinct metabolic characteristics of each model, enabling meaningful comparisons in our subsequent analyses.

Using this standardized dataset, with metabolic state (healthy or PDAC) as the class label, the random forest classification of the metabolic models achieved remarkable performance in distinguishing between healthy and cancerous states (S2 Fig). The model demonstrated high accuracy (94.83%) and perfect recall (1.0) for cancerous cases, ensuring no false negative classifications. Cross-validation results showed consistent performance across folds (mean CV score: $0.8214 \pm 0.0451$), with individual fold scores ranging from 0.7857 to 0.9310. The out-of-bag score of 0.8261 further confirmed the model's robustness. Among the misclassified instances, only three GAN-generated healthy models were incorrectly labeled as cancerous, indicating high specificity in identifying cancer-specific metabolic patterns.

## Systems-level analysis of metabolic reprogramming

Having validated our random forest model as an effective pattern recognition tool, we next leveraged this framework to perform a comprehensive systems-level analysis of the metabolic reprogramming patterns in PDAC (Fig 3). Rather than focusing on prediction, we utilized the model's feature importance metrics to systematically identify the most distinctive metabolic features that differentiate healthy and cancerous states. Through this pattern recognition approach, we discovered key reactions that are differentially regulated between healthy and cancerous metabolic profiles. The analysis

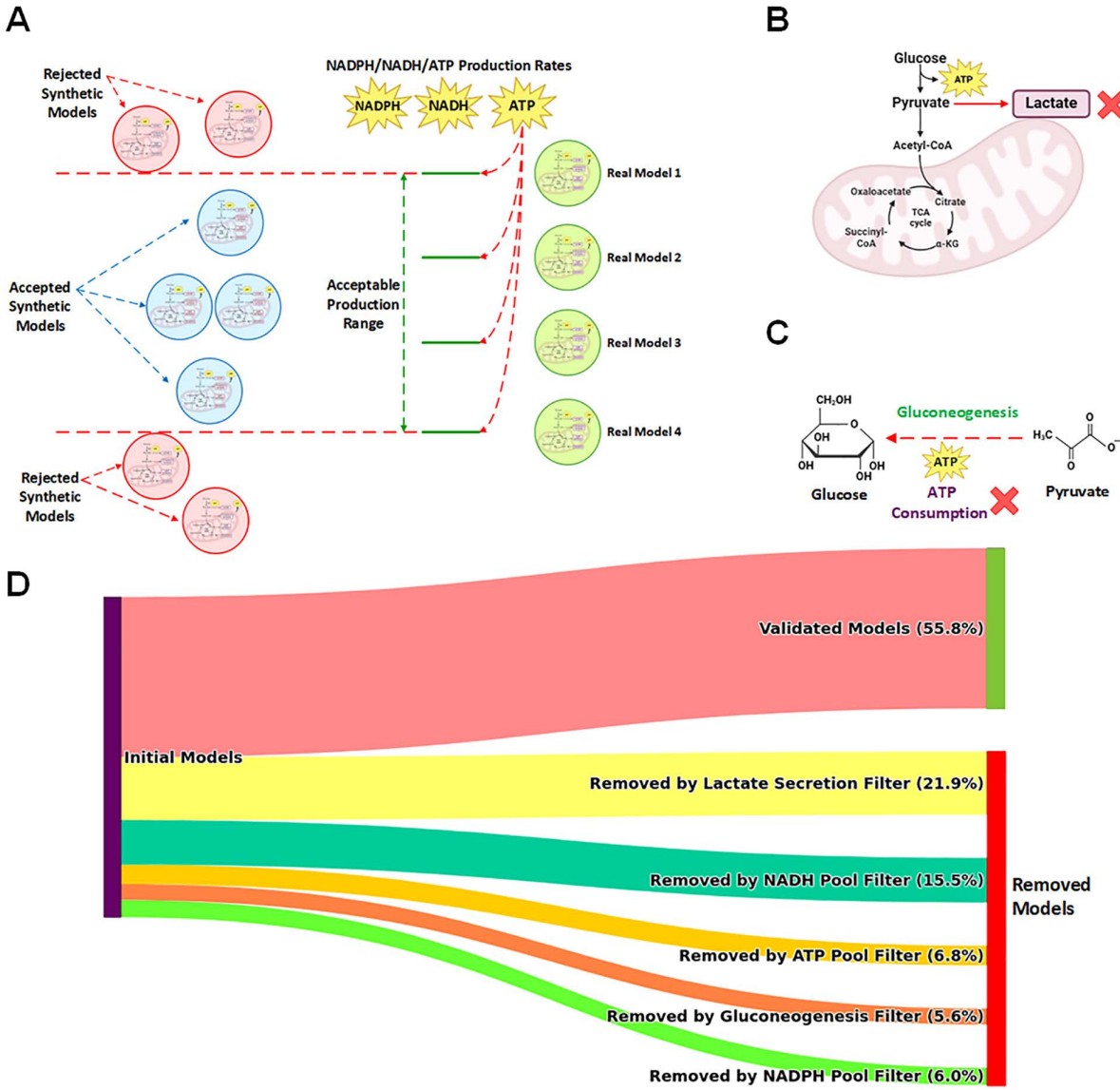

**Fig 2. Biological filtration process for evaluating GAN-generated metabolic models of healthy pancreas.** This figure illustrates the multi-step biological filtration process used to evaluate and refine GAN-generated metabolic models of healthy pancreas tissue. (A) ATP, NADH, and NADPH Production: Models were evaluated based on their production rates of ATP, NADH, and NADPH. Those falling outside the range established by four original healthy cases were removed. (B) Glycolysis Disorder Index (GDI): Models were assessed for their glucose metabolism patterns. Healthy models with a GDI equal to zero were retained, reflecting normal glycolytic function without the Warburg effect characteristic of cancer cells. (C) Gluconeogenesis: Models exhibiting gluconeogenesis, an ATP-consuming process typically absent in well-nourished cells, were excluded from the healthy dataset. (D) Filtration Flow: The diagram depicts the progression of input models through each filtration step, showing the percentage of models removed at each stage.

encompassed three levels of biological organization: reaction-level flux changes, pathway-level importance, and gene-level contributions to metabolic reprogramming.

Fig 3A illustrates the relationships between metabolic reactions and their associated pathways, with reactions grouped by hierarchical clustering (dendrogram on left) and color-coded by their respective pathways. The scatter plot displays the

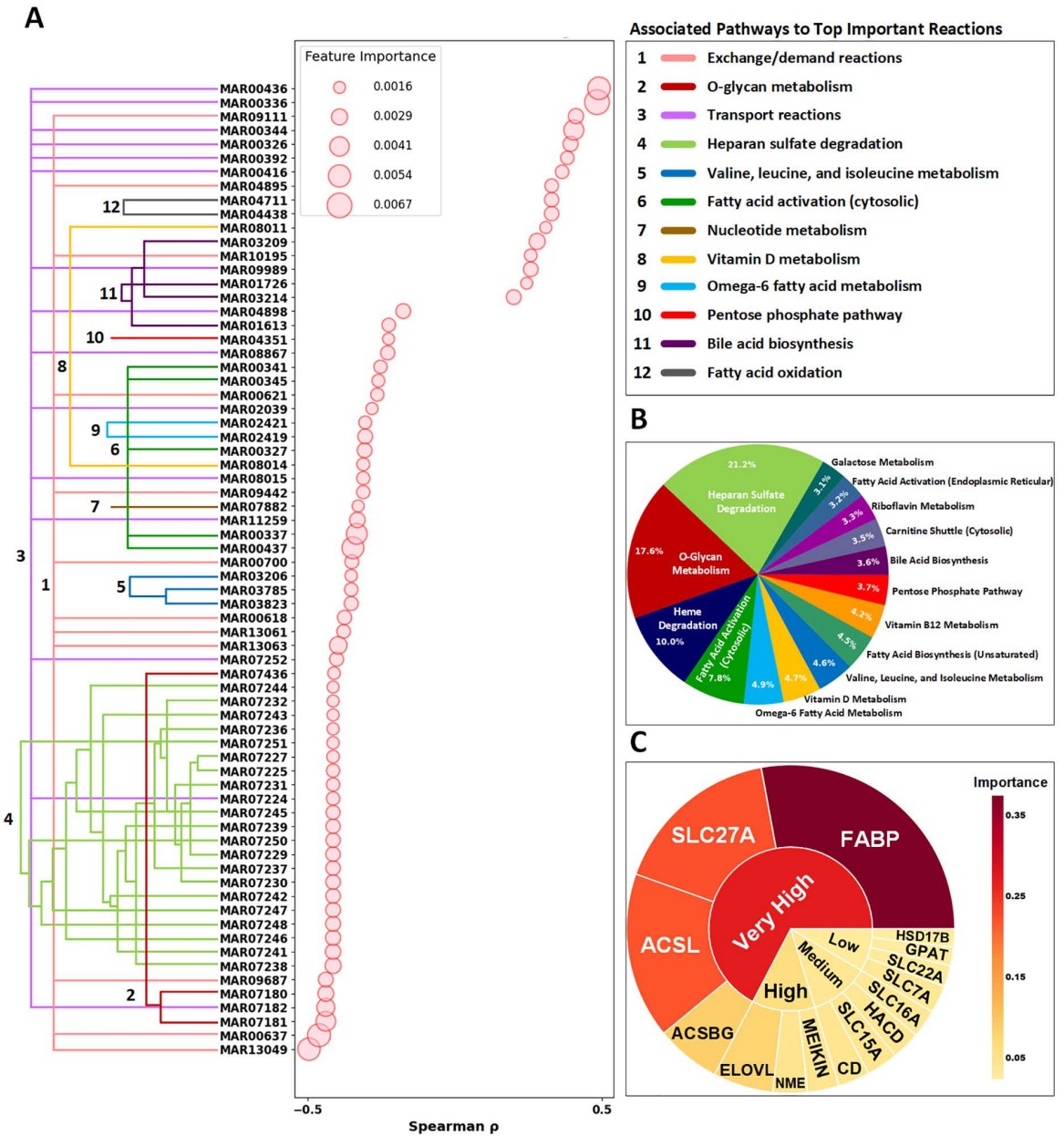

**Fig 3. Systems-level analysis of metabolic reprogramming in PDAC.** The figure presents a comprehensive analysis of metabolic dysregulation in PDAC using genome-scale metabolic models. (A) Reactions Importances and Their Correlations with Target (Top 70). Left panel shows a dendrogram grouping metabolic reactions with their corresponding pathways (colored branches), while the plot displays Spearman correlations ($\rho$) between reaction fluxes and disease state, with circle sizes proportional to feature importance scores from random forest analysis. (B) Pathway Importance (Top 15). Relative importance of metabolic pathways in distinguishing PDAC from healthy samples, with heparan sulfate degradation and O-glycan metabolism emerging as the most significant pathways. (C) Genes Families Importance (Top 15). Gene importance hierarchy derived from reaction-level feature importance scores, with genes categorized into "Very High," "High," "Medium," and "Low" importance based on their contribution to metabolic reprogramming in PDAC. For the "Very High" gene families: *FABP family includes: FABP1, FABP12, FABP2, FABP3, FABP4, FABP5, FABP6, FABP7, FABP9. SLC27A family includes: SLC27A1, SLC27A2, SLC27A3, SLC27A4, SLC27A5, SLC27A6. ACSLs family includes: ACSL1, ACSL3, ACSL4, ACSL5, ACSL6. ACSBGs family includes: ACSBG1, ACSBG2. The rest of names of the genes in each family are included in* S1 File.

Spearman correlation coefficients between reaction fluxes and disease state (healthy vs. cancer), with circle sizes proportional to feature importance scores derived from our random forest analysis. This visualization reveals both strongly positive and negative correlations with disease state, highlighting reactions that undergo significant flux alterations in PDAC. The most important reactions cluster predominantly in specific pathways, including heparan sulfate degradation, O-glycan metabolism, and fatty acid transport, which showed the strongest discriminative power between healthy and cancerous states.

For pathway-level analysis (Fig 3B), we calculated importance scores for each metabolic pathway by averaging the importance values of all reactions within that pathway. This approach identified heparan sulfate degradation, O-glycan metabolism, and heme degradation as the three most important pathways distinguishing between healthy and cancerous states. The relative contribution of each pathway reflects its overall importance in PDAC metabolic reprogramming, providing a systems-level view of the most altered metabolic functions.

At the gene level (Fig 3C), we utilized gene-protein-reaction (GPR) associations from the Human1 model to map each reaction to its corresponding genes. Using these associations, we calculated gene importance scores by aggregating the importance values of all reactions associated with each gene. To account for genes participating in multiple reactions, we implemented a sigmoid weighting function that gave additional weight to genes involved in numerous reactions while preserving the significance of genes with fewer but potentially critical roles. After obtaining importance scores for all gene families, we applied quartile-based thresholding to categorize the top 15 gene families into four tiers of importance ("Very High," "High," "Medium," and "Low"). This hierarchical organization revealed that genes encoding fatty acid binding proteins (FABPs), fatty acid transporters (SLC27As), long-chain acyl-CoA synthetases (ACSLs), and acyl-CoA synthetase bubblegum family members (ACSBGs) were the most significant drivers of metabolic differences between healthy and cancerous states, all falling into the "Very High" importance category.

This multi-level approach provided insights about the metabolic adaptations characterizing PDAC, connecting pathway-level dysregulation to specific reactions and their encoding genes. The integration of these three levels of analysis revealed coordinated patterns of metabolic reprogramming and identified potential vulnerabilities that could be targeted therapeutically.

**Key metabolic pathways in PDAC**

Our systems-level analysis identified several significantly altered metabolic pathways in PDAC, with heparan sulfate degradation (21.2%), O-glycan metabolism (17.4%), and heme degradation (10.0%) emerging as the three most differentially regulated pathways. Below, we present a detailed analysis of each pathway's dysregulation and its potential implications in PDAC pathogenesis.

**Heparan sulfate degradation pathway.** Pathway-level analysis revealed that heparan sulfate degradation was the most significant pathway discriminating between healthy and cancerous states, accounting for the highest importance score in our model (Fig 3B). To elucidate the underlying mechanisms of this finding, we performed reaction-level analysis within this pathway, focusing on reactions that our model identified as particularly important discriminators (Fig 3A). As shown in Fig 4A, probability density distributions comparing healthy and cancerous states reveal a consistent trend across all three key lysosomal degradation reactions (MAR07238, MAR07241, MAR07248). The overlaid density curves show healthy samples in blue and cancerous samples in red, with vertical dashed lines indicating mean flux values for each condition. Notably, all three reactions exhibited substantially lower activity in cancerous cells, as evidenced by the pronounced leftward shift of the cancer distribution (red) compared to the healthy distribution (blue) and the lower mean flux values (red dashed lines vs. blue dashed lines). These reactions are responsible for processing specific heparan sulfate degradation products (HS-Deg 13, HS-Deg 16, and HS-Deg 23) in lysosomes. The pronounced reduction in their activity explains why our model identified this pathway as the most critical factor distinguishing healthy from cancerous phenotypes. Importantly, these patterns are consistent when comparing cancer samples to the 4 real healthy samples directly, not just to synthetic samples (S1 Table), with fold-changes of similar magnitude, confirming that our findings are not artifacts of synthetic data generation.

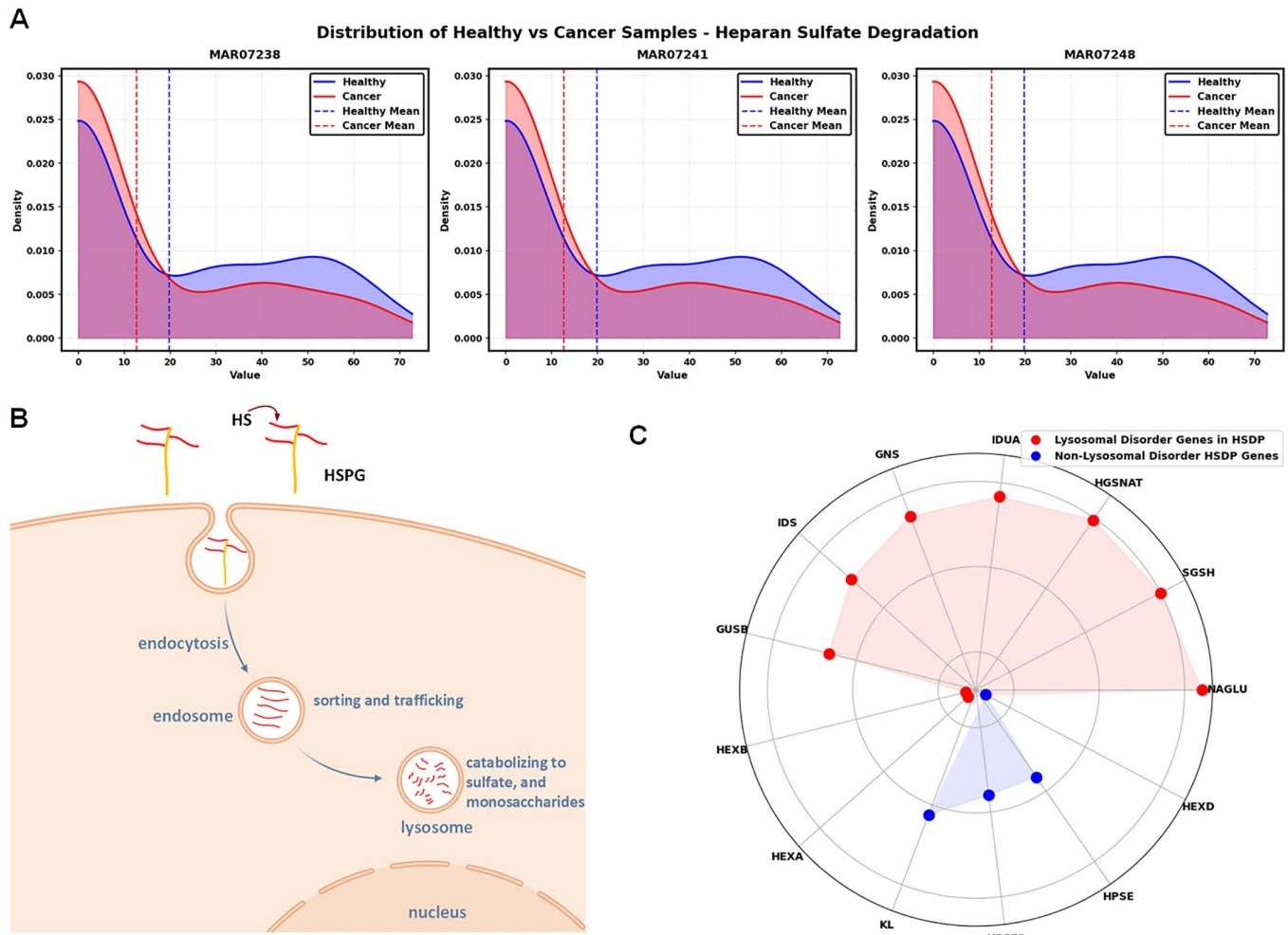

**Fig 4. Heparan sulfate degradation pathway and its implications in Pancreatic Ductal Adenocarcinoma (PDAC).** This figure illustrates the Heparan Sulfate Degradation pathway and its significance in PDAC, as revealed by genome-scale metabolic modeling and machine learning analyses. (A) Distribution of Heparan Sulfate Degradation Reaction Fluxes in Lysosomes between Healthy and Cancerous States. Probability density plots (Kernel Density Estimation) comparing the flux distributions of three key lysosomal degradation reactions (MAR07238, MAR07241, MAR07248) processing specific heparan sulfate degradation products (HS-Deg 13, HS-Deg 16, and HS-Deg 23) between healthy (blue) and cancerous (red) samples (n = 144 each). Vertical dashed lines indicate mean flux values for each condition. The consistently lower flux values and leftward shift of cancer distributions across all three reactions demonstrate reduced activity of the Heparan Sulfate Degradation pathway in PDAC, aligning with the observed decrease in HSPG exchange at the cell boundary. (B) HSPGs Journey from Cell surface to Lysosome: The journey of Heparan Sulfate Proteoglycans (HSPGs) from the cell surface to the lysosome is depicted, showing key steps in the degradation process. The pathway begins at the cell boundary with the exchange of HSPGs from the extracellular space, progresses through various intermediates, and culminates in the lysosome where degradation products are stored. Created in BioRender. Saha, R. (2025) https://BioRender.com/8uz6flm. (C) Gene Importances for Heparan Sulfate Degradation Pathway with Lysosomal Disorder Genes Highlighted in Red. The subfigure displays genes associated with the Heparan Sulfate Degradation pathway. Red dots indicate genes that overlap with those recently identified as contributors to lysosomal storage disorder in PDAC [37], while blue dots represent other pathway's genes. This overlap suggests a potential mechanistic link between Heparan Sulfate Degradation and the observed lysosomal storage disorder in PDAC.

In addition, our model pinpointed a key discriminating reaction in the exchange/demand reactions pathway. MAR09111, which regulates Heparan Sulfate Proteoglycan (HSPG) exchange at the cell boundary, was identified as a significant differentiator (Fig 3A). This reaction exhibited markedly reduced activity in cancerous cells, further supporting the role of disrupted heparan sulfate metabolism in pancreatic cancer progression. To contextualize these findings, it is important

to understand the biological significance of HSPGs. These macromolecules reside on the cell surface and participate in cell-cell and cell-extracellular matrix interactions, enzyme regulation, and multiple signaling pathways. Through these interactions, HSPGs regulate cell proliferation, survival, adhesion, migration, and differentiation, making them essential for normal cellular function [38]. As illustrated in Fig 4B, the HSPG degradation process begins at the cell surface, continues through endocytosis and partial degradation in endosomes, and culminates in the lysosome where final degradation products are generated and stored through reactions occurring at neutral pH.

It is well established that the tumor microenvironment in pancreatic cancer is characteristically acidic [39]. This acidity is a result of altered metabolism, hypoxia, and increased glycolysis in cancer cells [40]. Given that optimal heparan sulfate degradation requires a neutral pH environment, we hypothesize, based on our results, that the acidic conditions in PDAC may impair the efficiency of these degradation reactions, potentially explaining the consistently lower reaction rates we observed in cancerous states. The consistent lower activity across this degradation pathway suggests that flaws in this process could result in fewer or defective final degradation products in the lysosome. This observation aligns with recent literature demonstrating that genetic variants associated with lysosomal storage disorders are enriched in PDAC patients [37], suggesting lysosomal dysfunction may contribute to pancreatic cancer development. Our findings extend this knowledge by identifying heparan sulfate degradation as a specific contributor to this lysosomal dysfunction. As shown in Fig 4C, when examining genes coding for reactions in the heparan sulfate pathway, we found that the majority overlap with those reported to contribute to lysosomal storage dysfunction in PDAC (highlighted in red). Through our computational model, we have therefore identified heparan sulfate degradation pathway dysfunction as a potential mechanistic origin for the lysosomal storage disorders observed in pancreatic cancer.

**O-glycan metabolism.** O-glycan metabolism was identified as the second most significant pathway in our analysis (Fig 3B), with distinct flux distributions observed between healthy and cancerous states (Fig 5A). Reaction-level examination revealed substantial alterations in glycoprotein processing within the Golgi apparatus, particularly in four critical reactions (MAR07436, MAR07180, MAR07181, and MAR07182) involved in the sequential modification of serine/threonine residues. This is an evidence of disrupted glycosylation processes in cancer samples identified by our model. To understand these findings, it is essential to consider the biological basis of O-glycan metabolism. O-glycosylation is a post-translational modification process where sugar molecules are attached to the oxygen atoms of serine or threonine residues in proteins [41]. This pathway begins with the addition of N-acetylgalactosamine (GalNAc) to serine/threonine residues (MAR07436), forming the Tn antigen (Fig 5B). The Tn antigen serves as an intermediate structure that is typically further modified through sequential addition of sugars such as galactose and N-acetylglucosamine (GlcNAc), producing complex branched O-glycans [42].

Aberrant glycosylation in cancer disrupts O-glycan elongation, leading to the accumulation of truncated O-glycans, such as the Tn antigen, on epithelial cancer cell surfaces [42,43]. Analysis of reaction rates in Fig 5A reveals a marked reduction or near-complete loss of activity in the key enzymatic reactions responsible for Tn antigen elongation in pancreatic ductal adenocarcinoma (PDAC) compared to healthy controls. As a result, these truncated O-glycans fail to undergo proper processing and instead accumulate on the cell surface without further sugar modifications. This disruption, evident from our reaction rate analysis, provides a mechanistic explanation for the characteristic overexpression of truncated O-glycans in pancreatic cancer. Beyond identifying the cause of Tn antigen accumulation, our findings highlight how these truncated glycans alter cell signaling, adhesion, and immune interactions, ultimately promoting tumor progression. More importantly, our model pinpoints specific dysfunctional reactions (e.g., MAR07436-MAR07182), offering potential therapeutic targets to restore normal glycosylation and counteract cancer-associated glycan dysregulation.

**Heme degradation.** Analysis of the heme degradation pathway, the third most important discriminative pathway in our model, revealed significant metabolic alterations in PDAC (Fig 6A). Notably, the MAR11321 reaction displayed a reversed flux direction in cancer cases compared to healthy tissue, with cancer samples predominantly showing negative flux values. This metabolic shift leads to increased bilirubin production rather than its conjugation to bilirubin-monoglucuronide.

PLOS Computational Biology

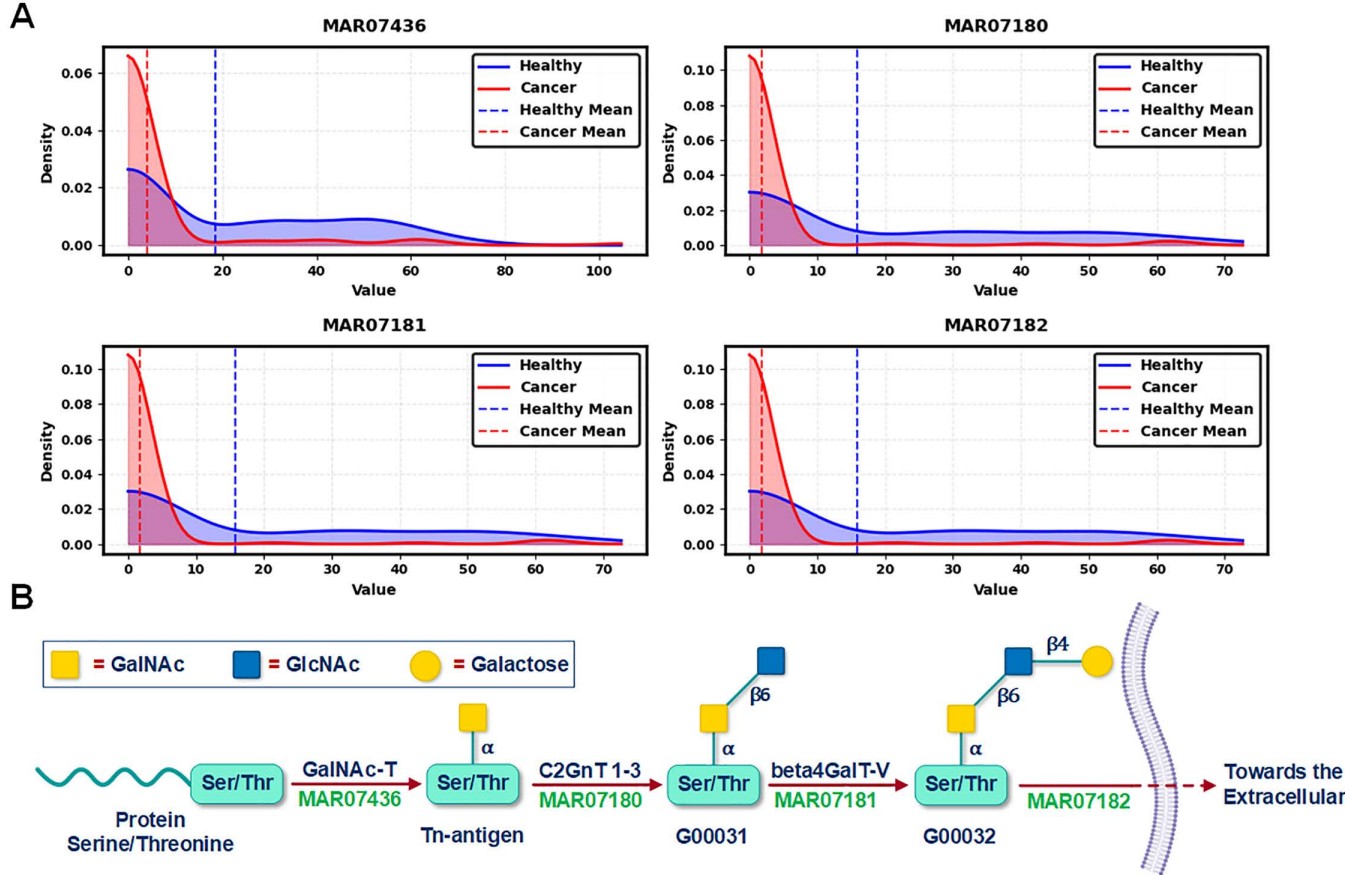

**Fig 5. Distribution of reaction fluxes in O-glycan metabolism pathway and their metabolic context.** (A) Reactions Fluxes Distribution for Healthy vs Cancer Samples. Probability density plots showing the distribution of reaction fluxes for four key reactions (MAR07436, MAR07180, MAR07181, and MAR07182) in O-glycan metabolism between healthy (blue) and cancer (red) samples (n = 144 each). The density values on the vertical axis represent the probability density function, indicating the relative likelihood of observing specific flux values, with higher peaks representing greater concentrations of samples around those values. Vertical dashed lines represent mean values for each condition. The total area under each density curve equals 1, providing a normalized representation of the flux value distribution in each metabolic state. (B) O-glycan Biosynthesis Pathway Schematic. Schematic representation of the O-glycan biosynthesis pathway occurring within the Golgi apparatus, showing the sequential modifications of Serine/Threonine (Ser/Thr) residues. The pathway begins with GalNAc-T-mediated addition of GalNAc to Ser/Thr residues (MAR07436) forming the Tn antigen, followed by sequential glycosyltransferase-mediated modifications (MAR07180-MAR07182) leading to complex O-glycan structures. The final reaction (MAR07182) facilitates the transport of the modified glycoprotein from the Golgi apparatus to the extracellular space. Yellow squares represent GalNAc residues, blue squares represent GlcNAc residues, and yellow circles represent galactose residues. The α and β annotations indicate the glycosidic linkage configurations. This simplified segment of the pathway shows the conversion of Tn-antigen to the more complex extracellular O-glycan structures, with each step corresponding to the flux distributions shown in panel (a).

Our finding provides a mechanistic explanation for the clinical observations of hyperbilirubinemia in 70–80% of patients with pancreatic head tumors [44].

To better understand the implications of this metabolic alteration, we examined the heme degradation pathway in greater detail (Fig 6B). This pathway follows a sequential breakdown process beginning with red blood cells, where hemoglobin is degraded to release heme. Heme is then converted to biliverdin through iron removal and porphyrin ring cleavage. Biliverdin is subsequently reduced to bilirubin [45], which normally undergoes glucuronidation via the MAR11321 reaction to form bilirubin-monoglucuronide, enhancing its water solubility for excretion. The reversed flux observed in

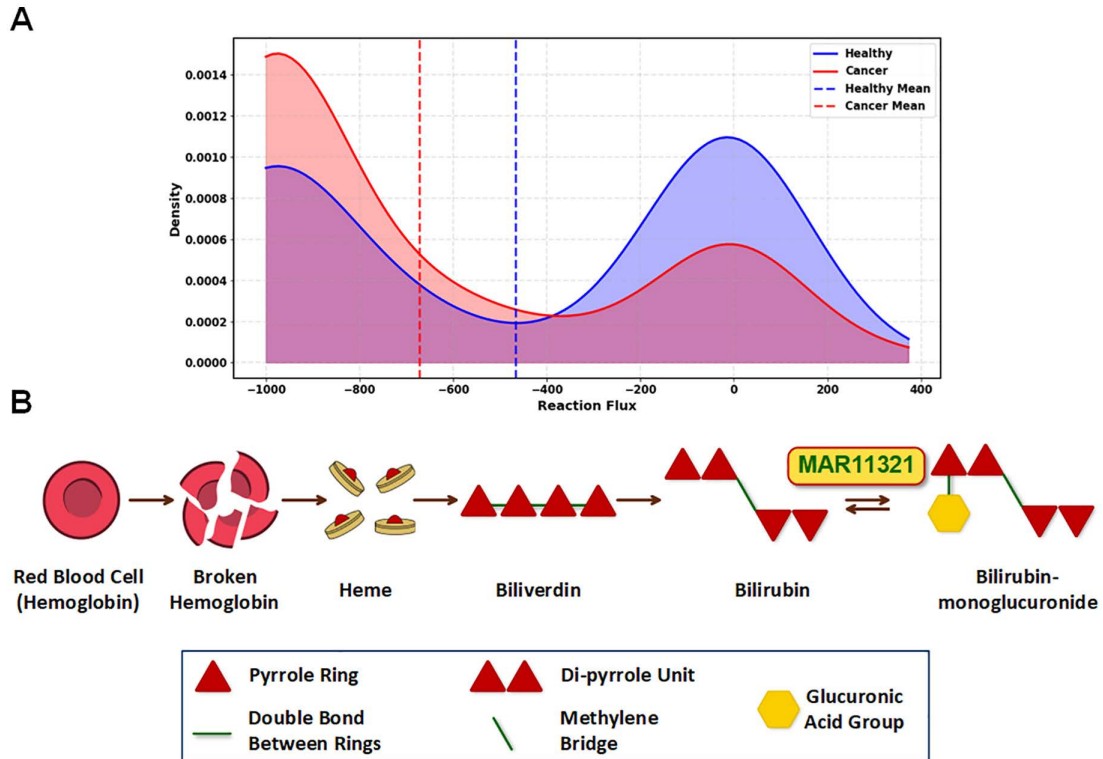

**Fig 6. Metabolic differences in the heme degradation pathway between healthy and cancer samples, with a specific focus on the MAR11321 reaction.** (A) Distribution of Healthy vs Cancer Samples for MAR11321 Reaction in the Heme Degradation Pathway. The density distribution plot demonstrates the reaction flux patterns across 144 healthy and 144 cancer samples. The cancer cases predominantly exhibit negative flux values, indicating a reversed reaction direction towards bilirubin production. This metabolic shift results in elevated bilirubin levels in cancer cases, particularly observed in pancreatic ductal adenocarcinoma where 70-80% of patients with pancreatic head tumors experience hyperbilirubinemia. (B) Heme Degradation Pathway Schematic. The pathway diagram schematically depicts the sequential breakdown of red blood cells, starting from hemoglobin degradation to heme, which is further converted to biliverdin and subsequently to bilirubin. The final step shows the conversion of bilirubin to bilirubin-monoglucuronide via the MAR11321 reaction.

PDAC suggests a disruption in this final conjugation step, leading to bilirubin accumulation—potentially contributing to the jaundice commonly seen in PDAC patients.

Beyond its role in jaundice, bilirubin accumulation may have broader implications for tumor progression and therapy resistance. Given its well-documented antioxidant properties, bilirubin could help cancer cells counteract oxidative stress, a mechanism frequently exploited by anti-cancer treatments (Nitti et al. 2020). Since hypoxia is a hallmark of the PDAC microenvironment, we speculate that the increased bilirubin production represents an adaptive strategy, allowing tumor cells to mitigate oxidative damage and enhance survival under these challenging conditions.

### Key metabolic reaction in PDAC

While pathway-level analysis provided insights into broad metabolic reprogramming in PDAC, examination of individual reactions revealed specific metabolic alterations at a finer resolution. Among all reactions analyzed, the nervonic acid transport reaction (MAR00336) emerged as the most significant reaction distinguishing healthy from cancerous states (Fig 7). Cancer samples exhibited consistently elevated transport activity and higher activation ratios (20–80%) compared to healthy samples, which showed only sporadic activation. This elevated transportation of nervonic acid in PDAC likely reflects a significant metabolic adaptation in this disease that is validated in real healthy samples (S1 Table).

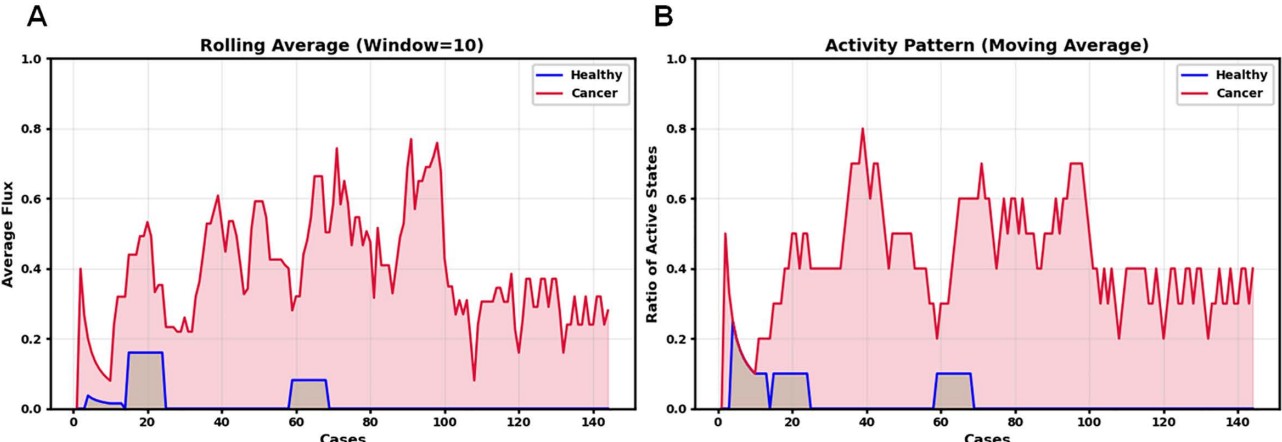

**Fig 7. Analysis of nervonic acid transport activity from extracellular space to cytosol in healthy and cancer samples.** (A) Rolling average flux values Comparison between Healthy and Cancer States. Rolling average flux values (window size = 10) demonstrates the metabolic flux intensity over sequential cases, revealing consistently elevated transport activity in cancer cases (red) compared to healthy cases (blue). The cancer samples maintain substantial flux values throughout the dataset, while healthy samples show minimal activity with only occasional peaks. (B) Activity Pattern of Reaction Fluxes for Healthy and Cancer Samples. Moving average of active states highlights the distinct patterns in transport activation between cancer and healthy cases. The activity is quantified as a ratio between 0 and 1, where 0 indicates complete inactivity (no flux) and 1 represents full activity (presence of flux) in all cases within the moving window. Cancer samples show consistent metabolic activity with activation ratios ranging from 20-80%, whereas healthy samples display sporadic activation confined to brief intervals. This differential pattern suggests a systematic upregulation of nervonic acid transport in cancer cells, which is consistent with altered lipid metabolism in cancer.

Considering nervonic acid's biological importance, this enhancement in its transport is particularly noteworthy. As a very long-chain omega-9 fatty acid (24:1), nervonic acid plays an essential role in membrane structure and function, particularly in sphingolipid composition [46]. PDAC cells undergo rapid proliferation requiring extensive membrane synthesis and remodeling, creating a heightened demand for specialized fatty acids like nervonic acid [47]. This increased need for membrane components is further amplified by PDAC's characteristic metabolic changes driven by KRAS mutations, present in most of the patients [48]. KRAS activates cellular pathways that increase the production and transport of fatty acids into cancer cells [49]. The consistent activation of nervonic acid transport observed in our cancer samples (20–80% activation ratio) versus the sporadic activation in healthy tissue suggests this is not merely a passive consequence of general metabolic upregulation but rather a specific adaptation that provides selective advantages to PDAC cells, potentially through optimized membrane composition supporting cancer cell growth and survival in the challenging tumor environment.

### Key metabolic genes in PDAC

Having characterized the key pathways and reactions in PDAC metabolism, we scaled our analysis down to the gene level to identify the molecular drivers behind these metabolic alterations. This gene-level analysis revealed several critical gene families driving the metabolic reprogramming in PDAC (Figs 8 and 3C).

Our model identified FABPs, SLC27As, ACSLs, and ACSBGs as the most important gene families distinguishing PDAC from healthy tissue (Fig 3C). Fig 8A illustrates how these gene families are associated with specific metabolic pathways, with the inner circle representing genes and the outer circles showing pathway associations. Color intensity reflects the relative importance of each element in discriminating between healthy and cancerous states.

The most important gene family, FABPs (Fatty Acid Binding Proteins), primarily functions in the transport reactions pathway. Notably, FABPs encode the nervonic acid transport reaction (MAR00336) that we previously identified as

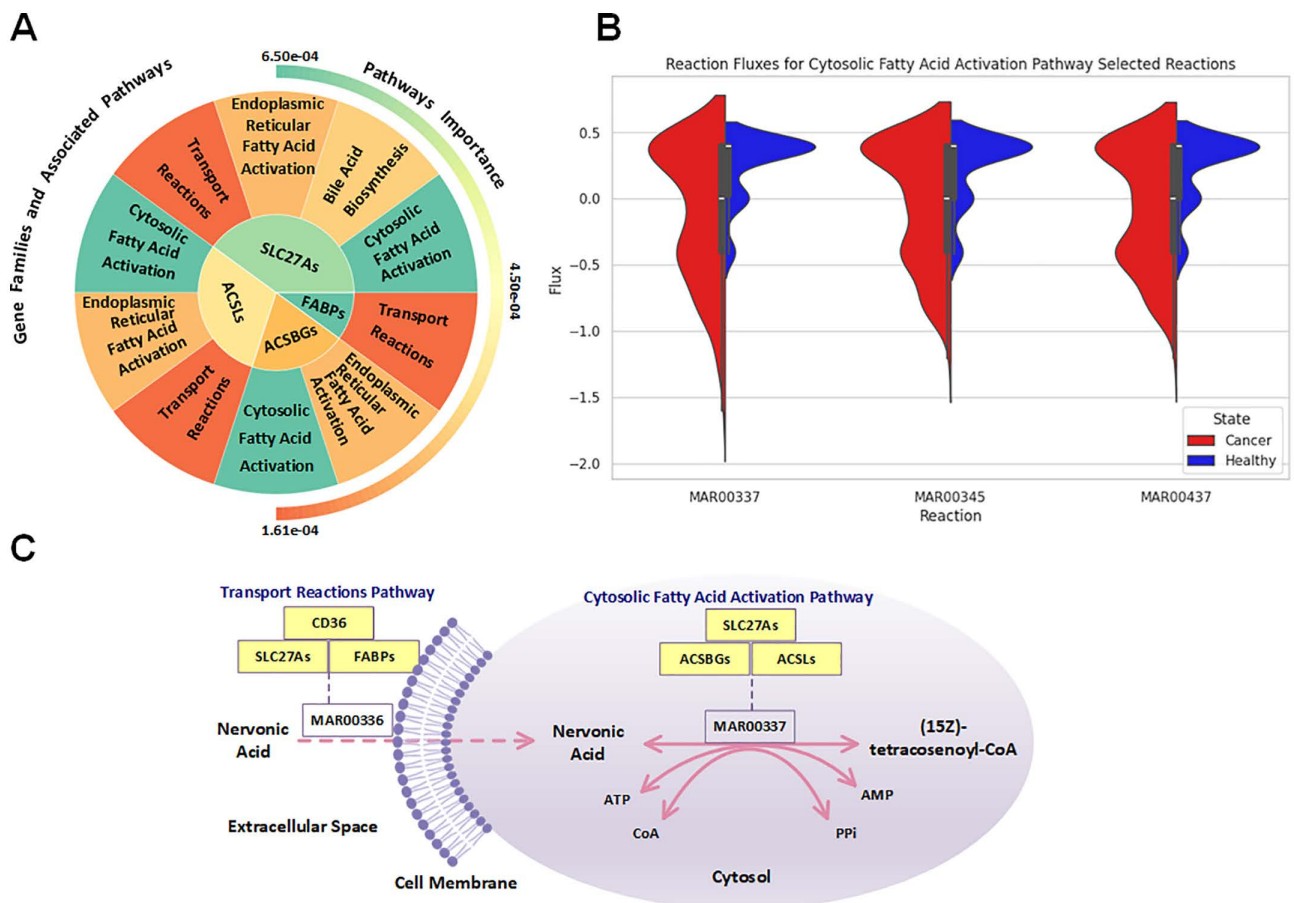

**Fig 8. Analysis of fatty acid metabolism and associated gene families in PDAC.** (A) Important Genes Identified by the Model and their Associated Pathways. Plot showing the relationship between key gene families (SLC27A and FABP) and their associated metabolic pathways, with the inner circle representing genes and outer circles showing pathway associations and their relative importance. (B) Fatty Acid Activation Pathway Selected Reactions Fluxes Distribution Comparison for Healthy vs Cancer Samples. Plots comparing reaction flux distributions between healthy (blue) and cancer (red) states for three key reactions (MAR00337, MAR00345, MAR00437) in the cytosolic fatty acid activation pathway, demonstrating consistently higher activation in cancer samples. (C) Schematic of Nervonic Acid Journey between Different Compartments and Pathways. Schematic representing nervonic acid path from extracellular to cytosol (from transport reactions pathway to cytosolic fatty acid activation pathway) while the reactions are being coded by the most important genes (refer to Fig 3C for the important genes).

the single most discriminating reaction between healthy and cancerous states. This provides a direct mechanistic link between our reaction-level and gene-level findings, as the same gene family mediates multiple fatty acid transport processes from the extracellular space to the cytosol.

To validate the importance of the second-ranked gene family, SLC27As, we examined reaction fluxes in the cytosolic fatty acid activation pathway where these genes play a crucial role. Fig 8B shows violin plots for three key reactions (MAR00337, MAR00345, MAR00437) in this pathway, demonstrating consistently higher activation in cancer samples (red) compared to healthy controls (blue). This increased activity confirms our model's prediction regarding the importance of SLC27As. Similarly, ACSLs and ACSBGs, which also encode enzymes in this pathway, show corresponding upregulation.

Fig 8C provides an integrative schematic that ties these findings together, illustrating how nervonic acid transport (mediated by FABPs and SLC27As) connects to subsequent activation in the cytosol (mediated by SLC27As, ACSLs, and

ACSBGs). This coordinated upregulation of both transport and activation processes represents a systematic reprogramming of fatty acid metabolism in PDAC. The increased activity of these pathways aligns with previous studies reporting enhanced cancer cell proliferation with FABP4 overexpression [50], underscoring the critical role of altered lipid metabolism in PDAC progression.

## Discussion

Our study presents a novel integrated approach combining genome-scale metabolic modeling with machine learning techniques to investigate metabolic reprogramming in pancreatic ductal adenocarcinoma (PDAC). This comprehensive analysis has revealed several key insights into PDAC metabolism and highlighted potential therapeutic targets.

The development of our GAN-based synthetic data generation method represents a significant advancement in addressing the persistent challenge of data imbalance in cancer research. Unlike previous studies that relied on traditional oversampling techniques or limited datasets, our approach generates biologically relevant synthetic samples while maintaining the complex relationships inherent in gene expression data. The rigorous three-step biological filtration process we implemented ensures the biological validity of synthetic samples, a crucial consideration often overlooked in previous studies using synthetic data generation methods. Our approach's uniqueness lies in its comprehensive validation framework, which extends beyond statistical metrics to include metabolic profiling, glycolysis disorder assessment, and gluconeogenesis evaluation. This multi-faceted validation strategy ensures that synthetic samples not only match the statistical properties of real samples but also reflect biologically meaningful metabolic states. The successful generation of 140 qualified synthetic healthy samples, validated through t-SNE visualization and metabolic analysis, demonstrates the robustness of our approach and its potential application in other cancer studies facing similar data imbalance challenges.

We next developed a machine learning framework on this balanced dataset of 144 healthy and 144 cancerous samples to identify metabolic reactions most altered in pancreatic cancer relative to the healthy state and used metabolic model-predicted reaction flux profiles from FBA as feature vectors—each reaction rate as a distinct feature—and applied random forest to classify healthy versus cancerous metabolic states. Our random forest classifier (94.83% accuracy) in distinguishing between healthy and cancerous metabolic states validates our approach to metabolic model comparison. Our innovative solution to the challenge of comparing models with varying reaction numbers through standardized padding represents a methodological advancement in metabolic model analysis. The perfect recall (1.0) for cancerous cases is particularly significant for clinical applications, as it minimizes the risk of false negatives in cancer detection.

Our subsequent multi-level analysis—spanning reactions, pathways, and genes—offers novel insight into PDAC metabolism, uncovering previously unrecognized links across distinct layers of metabolic dysregulation. To this end, Fig 3 serves as a powerful tool for understanding the complex metabolic landscape of PDAC. The dendrogram in Fig 3A effectively clusters metabolic reactions based on their associated pathways, illustrating how flux alterations can either uniformly affect all reactions within a pathway or selectively impact specific reactions. This clustering highlights functional relationships that extend beyond traditional pathway boundaries, revealing interconnected metabolic processes and emphasizing the complexity of metabolic reprogramming in pancreatic ductal adenocarcinoma (PDAC). The correlation coefficients and importance scores simultaneously displayed in this panel highlight reactions that not only differ significantly between healthy and cancerous states but also contribute substantially to the classification power of our model. This dual perspective prevents overemphasis on reactions that show large flux differences but may be biologically less relevant for distinguishing cancer from healthy tissue. The pathway importance analysis presented in Fig 3B transforms individual reaction-level findings into a more interpretable systems-level view. By revealing heparan sulfate degradation, O-glycan metabolism, and heme degradation as the top three pathways, this analysis directs attention to biological processes that might otherwise be overlooked in cancer metabolism studies, which traditionally focus on central carbon metabolism and energy production. The clear hierarchy of pathway importance provides a roadmap for prioritizing further investigation and potential therapeutic interventions. Fig 3C offers perhaps the most clinically relevant perspective by identifying specific

gene families that drive the observed metabolic differences. The prominence of fatty acid transport and activation genes (FABPs, SLC27As, ACSLs, and ACSBGs) highlights the critical role of lipid metabolism in PDAC, connecting our findings to the emerging recognition of fatty acid utilization as a key feature of pancreatic cancer metabolism. The organization of genes into importance tiers facilitates the prioritization of targets for functional validation and potential therapeutic development.

Overall, the integration of these three analysis levels creates a cohesive narrative of PDAC metabolism that would be impossible to achieve through any single perspective. By connecting specific gene activities to reaction fluxes and ultimately to pathway-level dysregulation, our approach bridges the gap between molecular alterations and functional metabolic consequences. While experimental validation is required to confirm these computational predictions, our framework provides a systematic, hypothesis-generating approach that prioritizes targets for laboratory investigation based on their discriminative power in metabolic modeling. It is important to note that reactions within metabolic pathways are often stoichiometrically or functionally coupled, meaning they must carry correlated fluxes. When our random forest model identifies multiple coupled reactions as important features, this reflects biologically meaningful coordination rather than statistical redundancy. For example, our identification of both fatty acid transport reactions (encoded by FABPs and SLC27As) and subsequent activation reactions (encoded by ACSLs and ACSBGs) as highly important features reveals that the entire fatty acid uptake and processing system is coordinately upregulated in PDAC (Fig 8C). Our multi-level analysis framework specifically addresses reaction coupling by: (1) aggregating reaction importances to the pathway level, which dampens the influence of individual coupled pairs while highlighting overall pathway dysregulation, (2) tracing coupled reactions back to their encoding genes to reveal mechanistic basis, and (3) using hierarchical clustering (Fig 3A) to explicitly visualize reaction groupings. This approach allows us to interpret coupled reactions as indicators of coordinated metabolic reprogramming—identifying entire metabolic modules rather than isolated reactions as dysregulated in cancer, which provides more actionable biological insights.

Among these, the identification of heparan sulfate degradation as the primary discriminating pathway between healthy and cancerous pancreatic tissue marks a substantial advance in our understanding of PDAC metabolism. The impairment we observed spans the entire pathway—from reduced HSPG exchange at the cellular boundary to diminished lysosomal degradation products—indicating systemic dysfunction rather than isolated enzymatic failure. This finding uniquely links extracellular matrix regulation with lysosomal processing, two cancer-related processes often studied in isolation. Our hypothesis that the acidic tumor microenvironment may impair lysosomal degradation through pH sensitivity offers a mechanistic explanation, and the overlap between key heparan sulfate genes and those implicated in lysosomal storage disorders further supports this connection. These insights nominate both heparan sulfate metabolism and lysosomal function as promising therapeutic targets.

The second most critical pathway, O-glycan metabolism, provides further mechanistic insight into PDAC-specific glycosylation patterns. While truncated O-glycans are well-documented in pancreatic tumors, our model pinpoints the precise metabolic disruptions responsible. Specifically, we identified severe reductions in four Golgi-localized elongation reactions (MAR07436, MAR07180, MAR07181, MAR07182), providing the first reaction-level explanation for the buildup of Tn antigens observed clinically. This distinguishes our work from previous studies that primarily described glycan structures without tracing their metabolic origins. By demonstrating that impaired elongation—not overproduction of precursors— drives this phenotype, we clarify the metabolic basis of an important cancer-associated feature and identify new targets within the glycosylation machinery for potential intervention.

Finally, our model uncovers a striking reversal of flux in heme degradation, particularly in reaction MAR11321, which catalyzes the conversion between bilirubin and bilirubin-monoglucuronide. While clinical jaundice has long been associated with pancreatic cancer, it is typically attributed to physical obstruction. Our findings suggest a deeper metabolic adaptation: cancer cells reverse this reaction, possibly to maintain intracellular bilirubin levels due to its antioxidant properties. This not only recontextualizes hyperbilirubinemia as a potential survival strategy under hypoxia but also implicates

bilirubin metabolism in therapy resistance. Targeting this specific reaction could disrupt an underappreciated redox-buffering mechanism, potentially sensitizing tumors to oxidative stress-inducing treatments.

Beyond the major dysregulated pathways identified, our analysis also highlights nervonic acid transport (MAR00336) as the most discriminative individual reaction between healthy and cancerous states. While lipid metabolism has been broadly implicated in pancreatic cancer, our model identifies this specific very long-chain fatty acid transport step as consistently active in PDAC (20–80% activation ratio) but largely absent in healthy tissue. This high-resolution insight moves beyond traditional pathway-level summaries, offering reaction-level granularity that can more directly guide therapeutic intervention. Importantly, nervonic acid plays a structural role in membrane composition and is involved in lipid-mediated signaling—both of which are critical for tumor growth and survival. By applying rolling averages and activation ratios across samples, we uncovered systematic upregulation patterns that may be missed by conventional analysis. The upregulation of this transport process may reflect adaptive membrane remodeling, potentially driven by oncogenic KRAS, thereby linking genetic drivers of PDAC to discrete metabolic dependencies and nominating a novel vulnerability for targeted therapy.

At the gene level, our model identifies FABPs, SLC27As, ACSLs, and ACSBGs as central molecular players orchestrating lipid metabolic reprogramming in PDAC. While isolated reports have noted altered expression of lipid metabolism genes in various cancers, our approach defines a functional hierarchy among these families and precisely maps them to reaction- and pathway-level changes that distinguish healthy from malignant tissue. This integrative mapping enhances our ability to identify not only which genes are dysregulated but how their activity translates into altered metabolic behavior across scales.

The prominence of FABPs in particular connects the molecular and functional dimensions of our model, directly linking gene-level upregulation with the increased flux through nervonic acid transport. This layered consistency—spanning gene, reaction, and pathway—reinforces the robustness of our findings and offers a multi-scale validation of this metabolic adaptation. Similarly, the consistent upregulation of cytosolic fatty acid activation enzymes encoded by SLC27As, ACSLs, and ACSBGs further supports the idea that PDAC cells strategically enhance their fatty acid processing capabilities. This metabolic shift likely supports increased demands for membrane synthesis, lipid signaling, and energy production, all of which are critical for cancer progression. Together, these insights suggest that fatty acid metabolism is not only rewired in PDAC but represents a coordinated and targetable feature of its metabolic phenotype.

Our integrated approach reveals a coordinated reprogramming of fatty acid metabolism that spans from extracellular transport to intracellular activation, suggesting these processes may be co-regulated in PDAC. This perspective advances beyond the typical focus on individual genes or pathways and offers a more complete picture of how multiple components of lipid metabolism work in concert to support cancer progression. Such mechanistic understanding could inform more effective strategies for metabolically targeted therapies that disrupt this coordinated system rather than individual components.

Our use of GAN-generated synthetic healthy samples addresses a critical challenge in cancer research—severe class imbalance in patient-matched study designs—but requires careful interpretation. Synthetic samples generated from only 4 real samples cannot capture the full biological diversity of healthy pancreatic tissue and are not statistically independent observations. While our rigorous three-step biological filtration (rejecting 44% of generated samples) ensures metabolic validity for tested characteristics, it cannot validate all biological constraints or gene regulatory relationships. Importantly, synthetic data may amplify idiosyncratic features of the original samples rather than universal healthy characteristics, and traditional statistical tests assuming independent samples may produce overconfident estimates.

We implemented several safeguards to mitigate these risks: multiple orthogonal biological validation criteria, t-SNE visualization confirming appropriate high-dimensional clustering, multi-level consistency checks across genes-reactions-pathways, and systematic comparison with established cancer biology literature. The biological concordance of our major findings (lysosomal dysfunction aligning with recent genetic studies, O-glycan patterns matching known cancer

glycosylation, lipid metabolism changes consistent with experimental validations) provides external validation that synthetic data did not introduce spurious patterns. Nevertheless, our results should be interpreted as hypothesis-generating computational predictions rather than definitive findings. The identified metabolic signatures require validation in independent cohorts of real samples and experimental confirmation through functional studies.

We emphasize that synthetic data cannot replace real data collection. Our approach represents a pragmatic solution for exploratory analysis and target prioritization when severe data scarcity exists, but the identified metabolic features must be validated experimentally before clinical application. Future studies should prioritize: (1) expanding real healthy pancreatic tissue biobanks, (2) multi-center collaborations to aggregate larger patient-matched cohorts, (3) experimental validation of predicted metabolic alterations through lipidomics, metabolomics, and functional assays, and (4) replication of findings in independent real datasets. The value of our framework lies not in providing definitive answers but in systematically prioritizing targets for such validation studies within the complex landscape of cancer metabolism.

While our integrated computational approach has yielded valuable insights, several important limitations should be acknowledged. First, the metabolic features identified through FBA represent computational predictions that require experimental validation before clinical application. Flux values serve as indicators of metabolic state differences but cannot be directly measured in patient samples. Future work should prioritize experimental validation of our key findings, particularly: (1) measurement of FABP, SLC27A, ACSL, and ACSBG expression levels in PDAC patient cohorts, (2) quantification of nervonic acid levels in tumors versus healthy tissue, (3) functional studies examining heparan sulfate degradation pathway activity, and (4) analysis of O-glycan profiles to confirm predicted truncation patterns.

Second, our FBA-based metabolic modeling has inherent limitations in capturing the full complexity of the tumor microenvironment. We employed uniform exchange reaction constraints across all samples, as patient-specific metabolomic measurements (oxygen tension, nutrient uptake rates, metabolite concentrations) were not available for the TCGA dataset. While this is standard practice when only transcriptomic data are available and enables identification of gene expression-driven metabolic differences, it does not fully capture the heterogeneous tumor microenvironment, including spatial oxygen and nutrient gradients, pH variations, or temporal metabolic dynamics. Cancer cells exist in spatially and temporally heterogeneous conditions with variable oxygen, nutrient, and growth factor availability that cannot be fully represented in computational models. However, our use of the iMAT algorithm with patient-specific gene expression data partially addresses this concern by constraining metabolic models based on the actual transcriptional state of each sample, which itself reflects adaptation to microenvironmental conditions. For instance, the gene expression data integrated through iMAT captures transcriptional adaptations such as upregulation of hypoxia-responsive genes, though explicit modeling of these factors would require additional experimental data. Additionally, our use of biomass maximization as the objective function, while extensively validated for proliferating cells, may not capture all metabolic objectives of cancer cells, such as maintaining redox balance, supporting invasiveness, or surviving under stress.

Third, the computational intensity of our approach may limit immediate clinical translation, suggesting the need for streamlined implementations or pre-computed signatures for clinical deployment. Future studies could integrate multiomics data (transcriptomics, metabolomics, proteomics), implement dynamic or multi-objective FBA approaches, and develop spatially-resolved metabolic models to better represent tumor heterogeneity.

Despite these limitations, our framework provides a systematic, hypothesis-generating approach that prioritizes targets for experimental investigation based on their discriminative power in comprehensive metabolic modeling. The biological concordance of our findings with established PDAC mechanisms and experimental literature (lysosomal storage disorders, truncated O-glycans, FABP overexpression) validates our modeling approach as capable of generating meaningful, testable hypotheses about cancer metabolism and suggests that experimental follow-up studies would be fruitful.

In conclusion, our integrated approach combining genome-scale metabolic modeling with machine learning has revealed novel insights into PDAC metabolism, identifying potential therapeutic targets and genes whose expression could serve as biomarkers upon experimental validation. The success of our methodology in addressing data imbalance

while maintaining biological relevance suggests its potential application in studying other cancers with limited healthy tissue samples. These findings not only advance our understanding of PDAC metabolism but also provide a framework for future studies combining metabolic modeling with machine learning approaches in cancer research.

## Materials and methods

### Data collection and preprocessing

We obtained gene expression data for pancreatic ductal adenocarcinoma (PDAC) from The Cancer Genome Atlas (TCGA) database. To ensure specificity, we meticulously reviewed annotations and pathology reports for 183 cases, selecting only those explicitly classified as ductal adenocarcinoma. This rigorous process yielded 144 PDAC cases and 4 non-neoplastic pancreatic tissue samples, which served as our control group. Importantly, these 4 healthy samples represent adjacent non-neoplastic tissue from the same patients who provided cancer samples, allowing for patient-matched comparisons that control for individual biological variation (age, sex, genetic background, race, and environmental factors). This paired tissue design is crucial for isolating cancer-specific metabolic changes, as using healthy samples from different individuals would introduce confounding biological variability that could obscure disease-specific alterations. Additionally, adjacent healthy tissue from the same patient experiences the same systemic metabolic microenvironment as the tumor (same blood supply, circulating factors, nutrient availability) while remaining phenotypically healthy, allowing us to investigate why some tissue resists malignant transformation despite exposure to the same patient-specific conditions—a question with direct implications for identifying resistance mechanisms and patient-specific therapeutic targets. While publicly available healthy pancreatic samples from other studies exist, incorporating them would introduce significant challenges: 1. most originate from adjacent tissue of patients with other pancreatic diseases rather than truly healthy individuals, as pancreatic resection is not performed without medical indication; 2. technical batch effects from different sequencing platforms and protocols could introduce artifacts particularly problematic for metabolic modeling, which is sensitive to relative expression levels; 3. loss of the patient-matched design would prevent identification of patient-specific metabolic vulnerabilities and obscure cancer-specific signals with population-level variation. Our approach of generating synthetic samples from patient-matched tissue addresses the common class imbalance problem in cancer machine learning research while preserving the biological and technical homogeneity essential for metabolic modeling. The dataset comprised Fragments Per Kilobase of transcript per Million mapped reads (FPKM) data from individuals with diverse ethnic backgrounds, ages, and sexes.

Initially, the dataset contained expression values for 60,616 genes across all samples. We applied a filtering step to remove genes with no read count across all samples, resulting in a refined set of 54,656 genes for subsequent analysis.

### Addressing data imbalance

To mitigate the significant imbalance between PDAC and control samples, we implemented a Generative Adversarial Network (GAN) approach to generate synthetic healthy gene expression profiles. Specifically, we adapted the Wasserstein GAN with Gradient Penalty (WGAN-GP) method from Kircher et al. [51] in their study on respiratory diseases of viral origin.

The WGAN-GP framework consists of two neural networks: a Generator and a Critic (Discriminator). The Generator produces synthetic data from a latent space, while the Critic attempts to distinguish between real and synthetic data. Through iterative training, the Generator learns to create increasingly realistic synthetic data, ultimately producing healthy gene expression profiles indistinguishable from the original samples.

Following Kircher et al.'s [51] implementation, we structured both the Generator and Critic networks as multi-layer perceptrons. The Generator consists of four layers:

- **Input Layer:** Accepts a 100-dimensional latent vector (z_dim).

- **Hidden Layers:** Three fully connected layers with dimensions $250 \rightarrow 500 \rightarrow 1000$ neurons, each followed by a LeakyReLU activation function ($\alpha = 0.2$).

- **Output Layer:** A fully connected layer with dimensions matching the number of genes in the dataset (10,755 features), using a Tanh activation function to scale outputs to [-1, 1].

  The **Critic (Discriminator)** consists of four layers:

- **Input Layer:** Accepts gene expression profiles (10,755 features).

- **Hidden Layers:** Three fully connected layers with dimensions $1000 \rightarrow 500 \rightarrow 250$ neurons, each followed by a LeakyReLU activation function ($\alpha = 0.2$).

- **Output Layer:** A single neuron for Wasserstein distance estimation without an activation function.

The model was trained for **2,000 epochs** with a **batch size of 2** and a **learning rate of 1e-4**, using Adam optimization ($\beta_1 = 0.0$, $\beta_2 = 0.9$). We implemented a **gradient penalty** ($\lambda = 10$) to enforce Lipschitz continuity by penalizing large gradients during training. This approach allowed us to augment our original dataset of healthy samples, effectively addressing the imbalance in our study.

## Qualification of synthetic data

We subjected the GAN-generated synthetic healthy samples to rigorous biological filtration to ensure their validity:

*Metabolic Profiling*: We established ranges for ATP, NADH, and NADPH production based on the four original healthy samples. Synthetic samples falling outside these ranges were excluded.

*Glycolysis Disorder Index (GDI)*: We calculated the GDI, defined as the ratio of lactate secretion to glucose uptake. In healthy cells, pyruvate primarily enters the TCA cycle rather than being converted to lactate. We retained only synthetic samples with a GDI of zero, removing models that did not meet this criterion.

$$Glycolysis\ Disorder\ Index\ (GDI) \ = \ \frac{Lactate\ Secretion}{Glucose\ Uptake}$$

*Gluconeogenesis Assessment*: We eliminated any synthetic healthy models exhibiting gluconeogenesis, as this process is typically inactive in well-nourished cells [52].

Following these stringent filtration steps, we obtained a balanced dataset comprising 144 PDAC samples and 140 qualified synthetic healthy samples plus 4 real healthy cases resulting in 144 control samples.

## Genome-scale metabolic modeling

We generated genome-scale metabolic models for all samples using the integrative Metabolic Analysis Tool (iMAT) within the COBRA Toolbox, employing the HumanGem model as a template. The HumanGem model includes eight cellular compartments (cytosol, mitochondria, endoplasmic reticulum, Golgi apparatus, lysosomes, peroxisomes, nucleus, and extracellular space) with reactions assigned based on enzyme localization and transport reactions connecting compartments [8]. First, genes were mapped to their corresponding reactions based on gene-protein-reaction (GPR) associations. Reaction expression levels were then calculated using these gene-to-reaction mapping function in COBRA toolbox, by using established GPR rules. Afterwards, the reaction expression levels of each sample were discretized into three levels - highly [1], moderately (0), and lowly expressed (-1) reactions. This discretization was based on a threshold: mean expression+0.3*standard deviation (SD) for highly expressed reactions and mean

expression− 0.3*SD for lowly expressed reactions. Reactions falling in between these thresholds were considered to be moderately expressed. The selection of 0.3 times the standard deviation as our threshold multiplier was based on its ability to produce a discretization pattern that approximates the distribution observed when using the upper 75th and lower 25th percentiles as cutoffs [21]. Subsequently, to reconstruct tissue-specific metabolic models for each sample, we employed iMAT method which is suitable for our biomarker discovery goals as it focuses on highly expressed genes and their associated reactions. This is because the highly expressed genes and reactions likely have a more significant impact on cellular functions and metabolic processes [53]. Finally, Flux Balance Analysis (FBA) was performed to obtain reaction velocity profiles for each sample, using biomass maximization as the objective function. This is a standard approach for modeling proliferating cells that has been extensively validated in cancer metabolism research [5,9,23]. The biomass reaction represents the production of cellular components (nucleotides, amino acids, lipids, etc.) required for cell growth and division, which is appropriate for the highly proliferative PDAC phenotype. Exchange reactions were constrained according to the default bounds in the Human1 model, representing physiological nutrient availability. These constraints were applied uniformly across all samples (healthy and cancerous), ensuring that metabolic differences identified by our models arise from gene expression-driven context-specific network structures rather than from pre-imposed constraint differences. While sample-specific metabolomic measurements (e.g., oxygen tension, nutrient uptake rates) were not available for the TCGA samples, the gene expression data integrated through iMAT captures cellular adaptation to microenvironmental conditions such as hypoxia, as evidenced by upregulation of hypoxia-responsive pathways in cancer samples.

## Machine learning approach

We implemented a Random Forest classifier to distinguish between PDAC and control samples based on their metabolic reaction profiles. The dataset was split into training (80%) and testing (20%) sets, maintaining class stratification. We utilized the scikit-learn library for model implementation and evaluation.

The Random Forest classifier was configured with 1,000 estimators, Gini impurity criterion, and balanced class weights. We performed 8-fold cross-validation and out-of-bag (OOB) score estimation to assess model performance and stability. The model achieved high accuracy on both training and test sets, with detailed performance metrics including precision, recall, and F1-score calculated for each class.

Feature importance analysis was conducted to identify the most discriminative metabolic reactions between PDAC and healthy samples. These reactions were then mapped to their corresponding pathways and genes, providing a multi-level systems perspective on PDAC-associated metabolic alterations.

## Supporting information

**S1 Fig. t-SNE visualization of gene expression data.** t-SNE plot showing the distribution of GAN-generated healthy samples (green), real healthy samples (blue), and cancer samples (red) in a reduced two-dimensional space. The GAN-generated healthy samples effectively follow the trajectory established by the original healthy samples, validating our generative approach and biological filtration process. This alignment confirms that our WGAN-GP model successfully learned the underlying distribution of healthy pancreatic tissue gene expression patterns. The cancer samples display a more dispersed, heterogeneous distribution that frequently overlaps with regions occupied by healthy samples, illustrating the inherent complexity of distinguishing between healthy and cancerous metabolic states based on gene expression alone.
(PDF)

**S2 Fig. Random forest classification performance.** Confusion matrix showing the classification performance of the random forest model in distinguishing between healthy and cancerous metabolic states. The model correctly identified 26

healthy and 29 cancerous cases, with only 3 healthy cases misclassified as cancerous and no false negative predictions, demonstrating high accuracy (94.83%) and perfect recall (1.0) for cancerous cases. This perfect recall for cancerous cases (29/29 correctly identified) is particularly significant in the context of cancer screening, where false negatives (missing cancer diagnoses) can have severe clinical consequences.
(PDF)

**S1 Table. Validation of key metabolic findings in real healthy samples.** Mean flux values for top-ranked metabolic reactions comparing real healthy samples (n = 4), synthetic healthy samples (n = 140), and cancer samples (n = 144). Fold-changes are calculated as cancer flux relative to real healthy flux. The consistency of directional changes and magnitude of fold-changes between cancer and both real and synthetic healthy samples confirms that identified metabolic signatures are not artifacts of synthetic data generation.
(PDF)

**S1 File. Complete gene family lists.** Complete lists of all genes in each gene family mentioned in Fig 3C, organized by importance tier.
(CSV)

## Acknowledgments

We gratefully acknowledge Dr. Adil Alsiyabi and Andrea Goertzen for their invaluable guidance and support. We also thank the High-Performance Computing Center (HCC) at the University of Nebraska-Lincoln for providing essential computational resources for this work.

## Author contributions

**Conceptualization:** Tahereh Razmpour, Masoud Tabibian, Rajib Saha.

**Data curation:** Tahereh Razmpour, Masoud Tabibian.

**Funding acquisition:** Rajib Saha.

**Investigation:** Tahereh Razmpour.

**Methodology:** Tahereh Razmpour, Masoud Tabibian, Arman Roohi.

**Project administration:** Rajib Saha.

**Software:** Tahereh Razmpour.

**Supervision:** Arman Roohi, Rajib Saha.

**Visualization:** Tahereh Razmpour, Masoud Tabibian.

**Writing – original draft:** Tahereh Razmpour, Masoud Tabibian.

**Writing – review & editing:** Tahereh Razmpour, Masoud Tabibian, Arman Roohi, Rajib Saha.

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
