## [Decision Letter · Decision Letter 0]

17 Sep 2025

GAN-Enhanced Machine Learning and Metabolic Modeling Identify Reprogramming in Pancreatic Cancer

PLOS Computational Biology

Dear Dr. Saha,

Thank you for submitting your manuscript to PLOS Computational Biology. The reviewers of this manuscript clearly appreciated the original approach taken in this study. After careful consideration, we feel that it has merit but does not fully meet PLOS Computational Biology's publication criteria as it currently stands. However, we feel that most points raised by the reviewers can be addressed. Therefore, we invite you to submit a revised version of the manuscript that addresses the points raised during the review process.

Please submit your revised manuscript within 60 days Nov 17 2025 11:59PM. If you will need more time than this to complete your revisions, please reply to this message or contact the journal office at ploscompbiol@plos.org. Please include the following items when submitting your revised manuscript:

We look forward to receiving your revised manuscript.

Kind regards,

Sunil Laxman, PhD

Academic Editor

PLOS Computational Biology

Marc Birtwistle

Section Editor

PLOS Computational Biology

**Additional Editor Comments:**

Reviewer #1:

Reviewer #2:

**Journal Requirements:**

At this stage, the following Authors/Authors require contributions: Tahereh Razmpour, Arman Roohi, Masoud Tabibian, and Rajib Saha. Please ensure that the full contributions of each author are acknowledged in the "Add/Edit/Remove Authors" section of our submission form.

2) We have noticed that you have uploaded Supporting Information files, but you have not included a list of legends. Please add a full list of legends for your Supporting Information files after the references list.

3) Some material included in your submission may be copyrighted. According to PLOSu2019s copyright policy, authors who use figures or other material (e.g., graphics, clipart, maps) from another author or copyright holder must demonstrate or obtain permission to publish this material under the Creative Commons Attribution 4.0 International (CC BY 4.0) License used by PLOS journals. Please closely review the details of PLOSu2019s copyright requirements here: PLOS Licenses and Copyright. If you need to request permissions from a copyright holder, you may use PLOS's Copyright Content Permission form.

Potential Copyright Issues:

i) Figures 1, 2, and 6. Please confirm whether you drew the images / clip-art within the figure panels by hand. If you did not draw the images, please provide (a) a link to the source of the images or icons and their license / terms of use; or (b) written permission from the copyright holder to publish the images or icons under our CC BY 4.0 license. Alternatively, you may replace the images with open source alternatives. See these open source resources you may use to replace images / clip-art:

4) Please amend your detailed Financial Disclosure statement. This is published with the article. It must therefore be completed in full sentences and contain the exact wording you wish to be published.

5) Thank you for stating 'All the codes and materials used for this study are available at https://github.com/ssbio/PDAC' Please note that, though access restrictions are acceptable now, your entire minimal dataset will need to be made freely accessible if your manuscript is accepted for publication. This policy applies to all data except where public deposition would breach compliance with the protocol approved by your research ethics board. If you are unable to adhere to our open data policy, please kindly revise your statement to explain your reasoning and we will seek the editor's input on an exemption.

**Reviewers' comments:**

Reviewer's Responses to Questions

**Comments to the Authors:**

Reviewer #1: This paper described novel GAN-Enhanced ML and metabolic modeling to study pancreatic cancer. The ML classifier achieved high accuracy in distinguishing between cell states and revealed key dysregulated pathways. The simulation provided mechanistic explanation for pancreatic cancer. The paper is well written and I have a few minor questions.

1. Biomarkers usually refer to unique metabolites or proteins. For cancer cells, the disease is due to genetic mutations. Therefore, the use of FBA to identify biomarkers is confusing to me. The flux value is determined by known multiple enzyme reactions as well as reaction thermodynamics. FBA may not have good resolution to reveal biomarker.

2. FBA flux is highly related to nutrient conditions. In cancel cell, the substrates can be complex and I am not sure if the FBA can simulate the realistic cellular status. What is the objective function for cancer cell metabolism.

3. The model discovered the disease is associated with fatty acid transporters and acyl-CoA synthetases that drive metabolic reprogramming. Some key metabolic reactions are controlled by cell organelles (such as mitochondria and peroxisome, etc.). It is unclear to me how FBA can sensitively predict cellular responses associated with compartmentalization.

4. Augmentation of realistic synthetic healthy samples (n=4) can be a bit risky. Augmentation techniques may lead models to learn patterns that do not generalize well to actual data.

5. lactate production in the cytosolic glycolysis pathway has been known as cancer- metabolism for many years. Can the model identify some new pathways that can be validated by recent experiments?

Reviewer #2: Summary:

This manuscript offers new computational insights into cellular metabolism with potential relevance to pancreatic cancer, specifically pancreatic ductal adenocarcinoma (PDAC). The authors analyze gene expression profiles from PDAC and healthy tissue samples, applying a computational pipeline that constructs tissue-specific metabolic models tailored to each sample. A notable innovation in this study is the application of a machine learning method—generative adversarial networks (GANs)—to address dataset imbalance. This enables the development of a random forest classifier that effectively distinguishes between cancerous and healthy states. The study further explores the metabolic reactions, pathways, and genes identified as key features by the classifier. Drawing on existing literature, the authors propose mechanistic hypotheses that differentiate PDAC metabolism from normal metabolism, highlighting roles for heparan sulfate, O-glycan biosynthesis, heme metabolism, and nervonic acid transport. These hypotheses point toward promising avenues for future mechanistic research.

Comments:

• In line 63, the manuscript states that late diagnosis and delayed treatment are major contributors to poor outcomes. Could the authors provide supporting evidence for this claim? Is it established that these factors are among the leading causes of mortality in PDAC? Clarifying this would reinforce the rationale for the study.

• The manuscript mentions a lack of gene expression data from healthy samples. Is this limitation specific to the dataset used in this study? Could publicly available data from healthy individuals in other studies be incorporated? What challenges would arise from doing so?

• Regarding the filtering of synthetic data, why were ATP, NADH, and NADPH production levels chosen as criteria? What makes these particular metabolites critical, and why were other currency metabolites not considered?

• More detail is needed on how flux balance analysis was conducted following the construction of tissue-specific metabolic models using iMAT. What objective function was employed? What constraints were applied to uptake and production rates? Given the discussion of hypoxia in PDAC, was oxygen uptake treated differently across sample groups?

• Could the authors elaborate on how inherent coupling between reaction fluxes—due to the structure of the metabolic network—might influence the random forest model? For instance, reactions that are tightly coupled across all samples may exhibit similar feature importance scores. How should this affect the interpretation of the model’s outputs?

• For Figure 4A, would a box plot or a probability density plot for each flux provide a clearer and more intuitive visualization of the results?

• Do the observed trends in reaction fluxes and pathway differences hold when analyzing only the real (non-synthetic) samples? While statistical significance may be limited due to the small number of healthy samples, does the real data support the same patterns?

• The study presents a method for analyzing gene expression data across conditions with unequal sample sizes by incorporating validated synthetic data generated via GANs. It would be helpful for the authors to discuss the risks, limitations, and necessary precautions when interpreting results derived from synthetic data.

• Line 152: The phrase should be revised to “deal with the class imbalance issue.”

**Have the authors made all data and (if applicable) computational code underlying the findings in their manuscript fully available?**

Reviewer #1: Yes

Reviewer #2: Yes

PLOS authors have the option to publish the peer review history of their article (what does this mean? ). If published, this will include your full peer review and any attached files.

**Do you want your identity to be public for this peer review?** For information about this choice, including consent withdrawal, please see our Privacy Policy .

Reviewer #1: No

Reviewer #2: No

**Figure resubmission:**
---

## [Decision Letter · Decision Letter 1]

21 Dec 2025

Dear Dr. Saha,

We are pleased to inform you that your manuscript 'GAN-Enhanced Machine Learning and Metabolic Modeling Identify Reprogramming in Pancreatic Cancer' has been provisionally accepted for publication in PLOS Computational Biology.

Best regards,

Sunil Laxman, PhD

Academic Editor

PLOS Computational Biology

Marc Birtwistle

Section Editor

PLOS Computational Biology

Reviewer's Responses to Questions

**Comments to the Authors:**

Reviewer #1: My comments have been addressed.

Reviewer #2: The authors have satisfactorily addressed the issues raised by the reviewer

**Have the authors made all data and (if applicable) computational code underlying the findings in their manuscript fully available?**

Reviewer #1: Yes

Reviewer #2: Yes

PLOS authors have the option to publish the peer review history of their article (what does this mean? ). If published, this will include your full peer review and any attached files.

**Do you want your identity to be public for this peer review?** For information about this choice, including consent withdrawal, please see our Privacy Policy .

Reviewer #1: No

Reviewer #2: No

---

## [Editor Report · Acceptance letter]

PCOMPBIOL-D-25-01528R1

GAN-Enhanced Machine Learning and Metabolic Modeling Identify Reprogramming in Pancreatic Cancer

Dear Dr Saha,

I am pleased to inform you that your manuscript has been formally accepted for publication in PLOS Computational Biology. Your manuscript is now with our production department and you will be notified of the publication date in due course.

With kind regards,

Anita Estes
